# Who Let the Dogs Out? Unmasking the Neglected: A Semi-Systematic Review on the Enduring Impact of Toxocariasis, a Prevalent Zoonotic Infection

**DOI:** 10.3390/ijerph20216972

**Published:** 2023-10-25

**Authors:** Katrin Henke, Sotirios Ntovas, Eleni Xourgia, Aristomenis K. Exadaktylos, Jolanta Klukowska-Rötzler, Mairi Ziaka

**Affiliations:** 1Department of Internal Medicine, Thun Hospital, Krankenhausstrasse 12, 3600 Thun, Switzerland; katrin.henke@spitalstsag.ch; 2Department of Emergency Medicine, Inselspital, University Hospital, University of Bern, 3008 Bern, Switzerland; sotirios.ntovas@insel.ch (S.N.); aristomenis.exadaktylos@insel.ch (A.K.E.); jolanta.klukowska-roetzler@insel.ch (J.K.-R.); 3Department of Visceral Surgery and Medicine, lnselspital, University Hospital, University of Bern, 3008 Bern, Switzerland; 4Department of Heart Surgery, lnselspital, University Hospital, University of Bern, 3008 Bern, Switzerland; elena.xourgia@gmail.com

**Keywords:** *Toxocara canis*, *Toxocara cati*, toxocariasis, parasitic infection, visceral larva migrans, *Albendazole*

## Abstract

Toxocariasis remains an important neglected parasitic infection representing one of the most common zoonotic infections caused by the parasite *Toxocara canis* or, less frequently, by *Toxocara cati*. The epidemiology of the disease is complex due to its transmission route by accidental ingestion of embryonated *Toxocara* eggs or larvae from tissues from domestic or wild paratenic hosts. Even though the World Health Organization and Centers for Disease Control classified toxocariasis amongst the top six parasitic infections of priority to public health, global epidemiological data regarding the relationship between seropositivity and toxocariasis is limited. Although the vast majority of the infected individuals remain asymptomatic or experience a mild disease, the infection is associated with important health and socioeconomic consequences, particularly in underprivileged, tropical, and subtropical areas. Toxocariasis is a disease with multiple clinical presentations, which are classified into five distinct forms: the classical visceral larva migrans, ocular toxocariasis, common toxocariasis, covert toxocariasis, and cerebral toxocariasis or neurotoxocariasis. Anthelmintic agents, for example, *albendazole* or *mebendazole*, are the recommended treatment, whereas a combination with topical or systemic corticosteroids for specific forms is suggested. Prevention strategies include educational programs, behavioral and hygienic changes, enhancement of the role of veterinarians, and anthelmintic regimens to control active infections.

## 1. Introduction

Human toxocariasis is an anthropozoonosis caused by the parasitic nematodes of the genus *Toxocara*, classified under the super-family of Ascaridoidea [1,2,3,4,5,6]. The term “toxocariasis” is used to designate the clinical spectrum of the human disease caused by the larval ascarids *Toxocara canis* (*T. canis*), the roundworm of dogs, and less frequently *Toxocara cati* (*T. cati*), the roundworm of cats [7]. Beaver et al. (1952) presented the first defined cases of human visceral toxocariasis [8]; meanwhile, the infection was acknowledged as a cosmopolitan disease reported in approximately 100 countries [7,9]. The reported seroprevalence of toxocariasis ranges between 2% and 37% (global seroprevalence rate of 19%) in urban and rural areas of Europe, Asia, and the USA, but may reach 85% in rural tropical regions [10,11,12,13,14,15,16,17,18,19,20,21,22,23,24]. 

In the natural definitive hosts (e.g., dogs and cats), the nematodes usually inhabit the digestive tract, from where they are expelled into the environment as eggs with feces [25,26], potentially spreading the infections to humans. Humans and paratenic host species can be infected directly by animals, or indirectly, that is, by ingestion of parasite eggs via consumption of contaminated vegetables and raw or undercooked meat from infected paratenic hosts [27,28]. After ingestion, juvenile larvae are encapsulated and hatch into the intestinal lumen, penetrate the intestinal wall, and migrate through the circulation in distal organs and systems, including the lungs, liver, eye, brain, and muscles leading to pleomorphic inflammatory and immune responses [29,30] and clinical presentations as well [31,32,33]. However, it is well-established that most human infections are asymptomatic or present non-specific signs and symptoms [34,35,36]. This last point, in combination with the unfamiliarity of the disease among health care providers, may explain the fact that although toxocariasis represents one of the most common helminthiases, the majority of the human infections remain undiagnosed, resulting in a potential underestimation of its zoonotic impact on public health [34,35,36,37]. 

Interestingly, in recent years, mounting evidence demonstrates a potential association of toxocariasis to chronic diseases such as asthma [38,39], and cognitive and behavioral disorders, including schizophrenia [40], neurodegenerative diseases [41], and epilepsy [42], making the investigation of the role of the pathophysiologic interactions between host and parasite in the pathogenesis of various systemic disorders a research challenge. This observation seems to be of utmost importance when one considers that helminthiases are the etiology for thousands of deaths and DALY (disability-adjusted life year) annually, and are responsible for a 6–35.3% loss in productivity [43,44,45]. 

This semi-systematic review aims to provide a comprehensive and critical overview of the existing literature on human toxocariasis, a neglected parasitic disease caused by the nematodes of the genus *Toxocara*. The review aims to synthesize and present key findings, concepts, and trends related to epidemiology as derived from relevant original research, narrative reviews and meta-analyses, pathophysiology, diagnosis, treatment, management, and prevention of toxocariasis. Additionally, the review will explore the potential association of toxocariasis with chronic diseases such as asthma, schizophrenia, neurodegenerative diseases, and epilepsy. Moreover, in the present work, we aim to strengthen clinicians’ awareness and propose less-explored topics as an objective for future research. In essence, the findings of this ongoing research are poised to provide valuable insights for public health authorities, aiding them in the implementation of more effective control strategies and informing the trajectory of future advancements within this field.

## 2. Methodology

A systematic literature search was conducted to identify relevant studies related to human toxocariasis. Study eligibility was assessed by three independent investigators (KH, SV, and EX) by review of article title and abstract and, when relevant, by text reading. In addition to PubMed and CENTRAL, we screened preprint servers (namely, medRxiv and Research Square) to capture rapidly accumulated evidence of the last year. We used the search phrase: (“toxocar*”). We adopted elements from both systematic and narrative reviews, leading us to term our approach as a ‘semi-systematic review’. While we followed rigorous and transparent procedures reminiscent of systematic reviews, such as defining a clear research question and utilizing systematic search strategies, we did not engage in the exhaustive and comprehensive literature search or the extensive quality assessment typical of full systematic reviews. On the other hand, unlike narrative reviews, our approach was more structured, and we aimed to minimize potential biases by integrating systematic methods where feasible. Our semi-systematic review provides a balance, ensuring the inclusion of relevant literature with a more manageable and focused scope, which was apt for the objectives of this paper. 

We retrieved relevant literature up to 15 August 2023, with no language restrictions. We considered for inclusion all manuscripts referencing *Toxocara canis* or *Toxocara cati* in regard to human disease. Moreover, experimental studies were considered if they provided valuable insights into the nature of toxocariasis. Furthermore, we have chosen to include a selected number of narrative reviews alongside primary research studies and systematic reviews and meta-analyses because they provide a historical context for the development of toxocariasis, aiding readers in understanding the evolution of research in this domain. Moreover, narrative reviews identify key themes, trends, and seminal works within the field and could serve as an instrumental in identifying gaps and controversies in the literature, which will inform the research questions addressed in this review. In addition, authored by experts in the field, narrative reviews provide synthesized interpretations, critical analysis, and expert perspectives that augment the overall understanding of the subject matter.

### Inclusion Criteria for Narrative Reviews

The selection of narrative reviews adheres to stringent inclusion criteria:

1. Relevance: Narrative reviews were considered eligible if they directly addressed aspects of toxocariasis and contributed insights relevant to our research objectives.

2. Author Expertise: We prioritized reviews authored by experts or recognized authorities in the field of toxocariasis to ensure credibility and reliability. To determine the authors’ expertise, we thoroughly reviewed the author’s academic and professional backgrounds, looking for relevant qualifications, research experience, and publications in the field of toxocariasis. We also considered their affiliations (i.e., academic institutions and research organizations), typically indicative of expertise. Authors’ previous work and contributions to the specific research field were additionally taken into account as further evidence of their expertise.

3. Quality Assessment: The selected narrative reviews were subjected to critical appraisal to evaluate their methodological rigor. Additionally, we assessed the criteria employed in selecting and excluding studies within the review. Furthermore, we thoroughly scrutinized potential biases, including but not limited to publication bias and selective reporting, and performed a detailed analysis of the measures taken to address these biases within the reviewed studies. Lastly, we conducted a comprehensive evaluation of the quality of evidence presented in the review, with a specific focus on the credibility of the sources and an overall assessment of the studies included in the analysis. Reviews found to have significant flaws were excluded.

Although our study generally excluded case reports and case series with less than ten patients, we made exceptions for 10 specific studies due to their unique relevance and significance.

The extracted data were organized based on key themes, including epidemiology, pathophysiology, clinical presentations, diagnosis, treatment approaches, management strategies, and preventive measures for human toxocariasis. Moreover, studies exploring the potential links between toxocariasis and chronic diseases were also summarized. Our study selection is summarized in Figure 1. 

## 3. Epidemiology

Toxocariasis represents one of the most common zoonotic helminth infections in industrialized countries [46], and is primarily caused by the ingestion of the eggs of nematode parasites of the genus *Toxocara*, commonly by the dog roundworm *T. canis* and less frequently by the cat roundworm *T. cati* [47,48].

The seroprevalence of toxocariasis varies worldwide [49]. Tropical and subtropical areas present the highest values—even higher than 50% [46,50]. Approximately 1.4 billion people are infected with or come into contact with *Toxocara* species worldwide [51]. In a meta-analysis by Rostrami et al. (2019), the global pooled seroprevalence of *Toxocara* was estimated at 19% (95% CI 16.6–21.4%). For Europe, its estimated impact appears to be around 18% [52]. However, despite its clinical impact being widely recognized, toxocariasis still represents a neglected disease, and major gaps in our understanding of the epidemiology of this parasite are present [53,54,55,56].

Infection tends to be more common in tropical regions due to the high humidity than in temperate regions, more common among rural populations with inadequate water supplies and poor housing than among urban populations, and more common in areas affected by poverty [57]. Feces of infected dogs make parks, town squares, playgrounds, sandboxes, and beaches the main source of human transmission. Moreover, reflecting children’s habit of geophagia and putting their fingers in their mouths during their playing activities is considered an infection of childhood [58,59].

Usually, these parasites are transmitted directly to the human host via the fecal–oral route and can cause toxocariasis and associated complications, including allergic and neurologic disorders. Humans can also become infected by ingesting encysted third-stage larvae in raw or inadequately cooked meat or organs from paratenic hosts, such as sheep, cattle, and chickens [60,61]. Consuming inadequately cooked or raw liver from infected animals has been associated with toxocariasis [60,61]. However, this way of transmission may be underestimated since transmission of *Toxocara* to humans via food consumption has not been the subject of extensive research [53] and should be further carefully evaluated. Moreover, despite the small number of embryonated eggs that can be found on the host hair of well-cared animals, human contact with contaminated hair of dogs or cats may represent a further way for transmission [60,61,62,63].

The life cycle of *Toxocara* spp. is characterized by its complexity (Figure 2) [64,65,66]. In dogs, usually under six months of age, the full life cycle of parasites includes a lung larval phase, followed by the development of adult worms in the small intestine. In contrast to young dogs, in adult dogs, adult worm development does not usually take place, and the parasites remain alive in various organs and tissues. Important for the parasite life cycle is the period of pregnancy in adult animals, as in this phase, larvae from various tissues migrate to the embryos resulting in the transplacental infection of puppies. Furthermore, dogs older than six months of age can acquire a patent infection by *T. canis*. This phenomenon occurs after infection with a low number of infected eggs or infection with larvae through the consumption of raw meat from infected paratenic hosts [65]. Subsequently, infected puppies lay immature eggs through their feces in the environment, resulting in environmental contamination [64,65,66,67,68,69]. This is followed by the process of maturation of the eggs into third-stage larvae inside them [65]. The eggs, which contain third-stage larvae, are accidentally ingested through contact with paratenic hosts, including humans (e.g., via contaminated food such as raw vegetables or fruit, water, or soil; (Figure 2) [64,65,66,70,71]. Although the conditions in the environment of the human intestine are not favorable at all for the development of adult parasites, the infective second-stage larvae that encapsulate from the eggs penetrate the small intestine and then travel with the systemic circulation and reach distant organs and tissues, where larval development ceases. However, despite that, the larvae migrate throughout the body, cannot mature, and instead encyst as second-stage larvae and can arrest in development for even years. [34,72,73]. It is well established that, primarily, the host’s inflammatory immune response to *Toxocara* is the pathophysiologic pillar of the pathology seen in toxocariasis, rather than mechanical damage associated with migrating larvae [74,75,76,77,78].

## 4. Pathophysiology

The causal relationship between eosinophils and helminth infection was first highlighted in the late 1800s by Müller and Rieder, who reported peripheral eosinophilia in two patients with hookworm infection [79]. Peripheral blood eosinophilia represents one of the main characteristics of helminthiases. Indeed, experimental mouse models infected with *Haemaphysalis longicornis* larvae develop significant peripheral blood eosinophilia, a finding confirmed in both animal and human studies of helminth infections, indicating the potentially protective role of eosinophils [80,81,82]. Moreover, eosinophil accumulation at the site of helminth infection has been described in multiple host species, ranging from zebrafish to humans [79,83]. Interestingly, eosinophils, which are cornerstones of the defense against parasitic infections, play a catalytic role in rejecting ticks in resistant hosts [80]. This observation is supported by experimental studies demonstrating that sensitized experimental animals against tick feeding are characterized by immune resistance to attacks and resistance, which is due to the increased accumulation of eosinophils around the embedded parts of the body [80,84].

Infection of the tissue triggers an inflammatory immune response in the body resulting in the clinical manifestations of toxocariasis (usually nonspecific), for example, fever, headache, cough, and pain [85,86]. It is well recognized that specific “pathogen-associated molecular patterns” exist for any helminth infection. For *T. canis*, these patterns are characterized by enhanced adaptive immune reactions and are primarily mediated by the CD4+ T-helper type 2 cell (Th2) [87,88,89], also called type 2 immune response. The elevated CD4+ Th2 activity leads to the production and release of inflammatory mediators—especially interleukin (IL)-4, -5, -10, and -13—triggering the production of immunoglobulin (Ig)E antibodies and the differentiation of eosinophils [88,90,91]. Experimental studies in mice have demonstrated a biphasic response with two peaks: an early peak at day ten and a late peak at day 21. Interestingly, in CD4^+^ T cell-deficient mice, only the early peak remains intact, and the late peak is absent, leading to the conclusion that it is generated by the innate rather than the adaptive Type-2 response [92]. Moreover, tissue damage caused by invasive parasites can release host-derived alarmins, such as IL-33 and IL-25, initiating a Th2 inflammatory response. Additionally, molecules like ADP and chemosensory tuft cells in the intestinal tract can contribute to eosinophil recruitment and activation. Parasite-derived molecules also play a role in modulating eosinophil response. Once the infection is established, eosinophils accumulate near parasites or sites of tissue damage, with CCL11 (eotaxin-1) and its receptor CCR3 being important in this process [79]. However, a better understanding of the role of molecular mechanisms in the pathogenesis of helminth infections is urgently needed. For example, regarding other helminthiases, recently, a significant pathophysiological role appears to be attributed to longistatin, a soluble salivary gland protein with two functional EF-binding Ca2+ motifs from the tick *H. longicornis* [44,80], which is secreted and injected into host tissues via saliva at the sample point. It appears that longistatin can modify the host’s inflammatory responses, as it has been shown that there is a massive influx of inflammatory cells at the tick sample site in the presence of longistatin [80,93,94]. 

The pathophysiology of eosinophil-associated tissue and organ damage is complex. Isolated eosinophil accumulation cannot be the only explanation. Various clinical and experimental studies highlight the considerable contribution of pro-inflammatory and tissue-damaging properties of eosinophils to the pathogenesis of many eosinophil-related diseases and hypereosinophilic syndromes [95]. Indeed, eosinophils produce various deleterious products, for example, basic proteins, superoxides, and free radicals [96], contributing to tissue damage, repair, remodeling, and disease persistence [97]. The cardinal involvement of eosinophils in the development of tissue remodeling and fibrosis is mediated through remodeling and fibrogenic growth factors, such as IL-3, IL-5, and the granulocyte-macrophage colony-stimulating factor (GM-CSF) [98,99,100]. More specifically, an enhanced production of eosinophilopoietic cytokines and growth factors, including IL-3, GM-CSF, and IL-5, leads to increased differentiation and uncontrolled proliferation. Subsequently, this magnifies migration, adhesion, and activation of the eosinophils [95,101]. Moreover, the intensive monoclonal proliferation of eosinophils from myeloid progenitor cells related to gene reorganization of oncogenic tyrosine kinase receptors may also play a fundamental role in the pathophysiology of hypereosinophilia and hypereosinophilic syndrome [102]. It is highlighted, however, that the eosinophils of infected individuals are ineffective in killing the parasite [103], with experimental studies demonstrating that *T. canis* shows an enhanced resistance against eosinophils [96]. Indeed, even though an occasional mass elevation of peripheral and tissue eosinophil levels is seen in toxocariasis, experimental studies of *T. canis* consistently demonstrate no effect of eosinophil depletion on worm killing. Moreover, the numbers of larvae recovered from host tissues were not altered in primary infected IL-5 transgenic mice [96,104] or IL-5 knock-out mice [105]. Consistent findings were also observed following the depletion of eosinophils with anti-IL-5 antibodies following secondary challenge [106].

Interestingly, the excretory–secretory products generated by the parasitic organism *T. canis*, denoted as TcE and TcEFs, may contain constituents endowed with anti-inflammatory properties, capable of suppressing the activation of the deleterious immune responses that underlie inflammatory diseases. Specifically, fractions of these products have demonstrated the ability to elicit the production of immunoregulatory cytokines, including TGF-β and/or IL-10, as well as proinflammatory IL-12 and Th1 cytokines, such as TNF-α. Notably, these fractions do not appear to stimulate the production of Th2 cytokines, such as IL-5 and IL-13, which are typically associated with allergic and Th2-mediated immune responses, nor do they induce the production of IL-17 cytokines. Further investigation is warranted to scrutinize these specific fractions in depth, with the aim of identifying potential candidate molecules suitable for the development of immunotherapeutic vaccines designed to mitigate the impact of inflammatory and autoimmune diseases [107].

Visceral larva migrans (VLM) is the most common syndrome in infected individuals, especially children, with clinical signs including cough, myalgias, or skin manifestations (e.g., pruritus, eczema, panniculitis, and vasculitis [108,109]. Migrating larvae cause a host immune response, resulting in local inflammation associated with eosinophilia and an increased production of cytokines and specific antibodies [17,85,110].

Furthermore, cardiac symptoms seem to be generated from the combination of direct larva invasion and enhanced immunological/hypersensitivity reactions—particularly associated with hypereosinophilia or immune responses induced by the presence of the parasites in tissues. The release of antigens from dying helminths can enhance these reactions during specific therapy with anthelmintics [111,112,113,114], making steroids a well-considerable complementary therapy.

Regarding neurotoxocariasis (NT), various factors have been suggested to contribute to its pathogenesis, including the above-mentioned T helper type 2-driven immune response with subsequent production and release of IL-5 and IgE [20,115]. Furthermore, studies utilizing immunohistochemistry to examine the cellular structure of microglia/macrophages and identify the presence of accumulated β-amyloid precursor protein (β-APP) as a sign of axonal harm in experimentally *T. canis*- and *T. cati*-infected mice demonstrated a notably elevated occurrence of demyelination and/or gitter cells within the cerebrum of the *T. canis* group compared to the *T. cati* group [116]. Moreover, further well-recognized mechanisms refer to the development of vasculitis, secondary to endothelial injury caused by cationic proteins released by eosinophils and IL-6 and the formation of immune complexes [117,118,119,120,121]. Furthermore, similarly to the pathophysiology of cardiac lesions, the release of helminth antigens—particularly from dying helminths during therapy [112]—seems to play a cardinal role [118,122]. In addition, the release of major basic protein and eosinophilic cationic protein and the production of anticardiolipin antibodies, associated with chronic inflammation, significantly contribute to the development of ischaemic lesions observed in this specific patient population [78,123,124]. Ultimately, experimental studies of *Toxocara* spp. infection underscore the profound influence of *Toxocara* spp. on the oxylipins—an essential class of bioactive regulatory lipids—within the intricate molecular signaling framework of infection, inflammation, and immune response in paratenic hosts [116]. Recognition and understanding of the associated pathophysiological mechanisms may improve clinical outcomes and therapeutic implications.

## 5. Clinical Presentations

Historically, the first recorded mention of *T. canis* dates back to 1782 when the German Parasitologist Paul Christian Friedrich Werner announced the discovery of this parasite as *Lumbricus canis* [125]. However, the recognition of the clinical importance took place many years after the first description, when Wilder discovered the presence of nematode larvae in removed enucleated eyes from children who were falsely diagnosed with retinoblastoma [126]. The discovery of the nematode stimulated the identification of the parasite *T. canis* by Beaver et al. [8]. Indeed, even though migrating ascarid larvae in human tissues were described during the first third of the 20th century [127,128,129], the definition of visceral toxocariasis as a clinical syndrome occurred in 1952 by Beaver et al. when the authors reported three patients suffering from various clinical manifestations including fever, anorexia, hepatomegaly, cough, eosinophilia, and anaemia [8]. Five years later, *T. canis* was isolated in a liver biopsy of one of the patients, and the disease was defined as VLM syndrome [130]. Currently, human toxocariasis is classified into five clinical forms: classical VLM, OT, common toxocariasis (comT), covert toxocariasis (covT), and cerebral toxocariasis or NT. The possibility of symptomatic disease, its type, and its severity is associated with the parasite load, the duration of parasite migration, the migratory pathway, and the parasite-induced host immune responses [7,34,35,131,132,133].

### 5.1. Visceral Larva Migrans

Although visceral toxocariasis is a zoonotic disease of childhood primarily affecting children between 2 and 7 years old, it is documented that the proportion of individuals with antibodies to *Toxocara* species increases with age. Infections in adults have been described worldwide and are mainly caused by ingesting raw meat [71,134,135].

The clinical board of visceral toxocariasis is highly pleomorphic, affecting several organs and systems, including the liver, heart, lung, and skin, with the liver being described as the most affected organ [39,73,112,136,137,138,139,140,141,142,143,144,145,146]. The most common presenting symptoms include a combination of nonspecific manifestations, for example, fever, anorexia, fatigue, abdominal pain, and lymphadenopathy [39,73,112,136,137,138,139,140,141,142,143,144,145,146]. Hepatic toxocariasis is characterized by the formation of granulomatous nodules and hepatitis [39,73,112,136,137,138,139,140,141,142,143,144,145,146]. VLM hepatic nodules are described as small, oval, or elongated non-spherical lesions with fuzzy margins and absent or insignificant rim enhancement on contrast-enhanced computer tomography (CT) scans [147,148]. Moreover, they may express cystic characteristics, making early and correct diagnosis difficult [149,150]. Indeed, it is reported that toxocariasis-associated hepatic lesions may be misinterpreted as tumors or pyogenic abscesses [147,150,151].

Cardiac involvement represents an extremely rare but occasionally fatal manifestation of visceral toxocariasis [112,139]. The severity of the illness ranges from asymptomatic to severe and potentially lethal. The reported toxocariasis-associated cardiac presentations include myocarditis, heart failure, endocarditis and endocardial thrombosis, and pericarditis up to cardiac tamponade [112].

Pulmonary toxocariasis occurs when the parasites migrate to the lungs and is associated with various symptoms, such as cough, dyspnea, and wheezing [138,152]. In the majority of the cases, the lung lesions involve more than three lobes with predilection of the subpleural region and the lower lung zone. Morphologically, there are two described radiologic patterns, ground-glass opacities (GGOs), solid nodules, consolidations, and linear opacities [138]. Rarely, toxocariasis may manifest as Loefler’s syndrome, characterized by pulmonary infiltrates and associated with peripheral eosinophilia [153]. Finally, the association between *Toxocara* species seropositivity and asthma has been widely discussed, with several studies indicating a pathophysiological relationship contributing to the development of asthmatic symptoms [39,154,155,156,157,158,159,160,161]. A meta-analysis of ten studies with 1530 participants highlighted a significantly higher incidence of *T. canis* infection in asthmatic individuals [162].

The association of human toxocariasis to skin disorders has been highlighted in various publications. The most prevalent cutaneous manifestation associated with human toxocariasis is chronic urticaria. Other common dermatologic entities of the disease include chronic pruritus, atopic dermatitis, and miscellaneous eczema in patients with *Toxocara* antibodies. Notably, isolated cutaneous toxocariasis without systemic symptoms may also occur, making the disease a diagnostic challenge [73,163]. Interestingly, toxocariasis has been reported to cause Reiter syndrome in a patient with chronic erythematosquamous balanitis and recurrent sternocostal arthralgia. This observation, however, represents an extreme scarcity [164].

Laboratory findings of visceral toxocariasis include leucocytosis, eosinophilia, anaemia, and hyperglobulinaemia of IgM, IgE, and IgG [8,165,166]. In addition, enhanced titers of isohemagluttinin A and B can be found [167]. 

### 5.2. Ocular Toxocarosis

OT is a syndrome mainly affecting older children, adolescents, and adults, and shows clinical diversity, partly reflecting the age of the affected individual [37,168,169,170,171]. Ocular disease is associated with the migration of the parasites through the circulation into the posterior eye segment with subsequent local release of inflammatory responses [152,169]. The most prevalent form of the disease is the development of peripheral granuloma of the eye followed by granuloma of the posterior pole of the eye, commonly extending from the macular landscape to the central retinal periphery and always accompanied by vitritis [170]. The disease occurs more frequently unilaterally; however, bilateral ocular infestation has also been described [169]. The clinical manifestations include disturbed vision, ablatio of the macula, and heterotopia [103,172,173,174,175,176,177,178,179,180,181,182,183,184]. Children can also develop leukocoria, strabismus, photophobia, eye pain, or eye redness [37]. Moreover, in pediatric patients, OT may also be manifested as optic nerve neuropathy, fulminant endophthalmitis, or papillitis with or without glaucoma, and rarely as diffuse chorioretinitis due to migration of dying larvae [30,170].

### 5.3. Neurotoxocariasis or Cerebral Toxocariasis

The invasion of *T. canis* through the blood–brain barrier into the CNS (i.e., brain and spinal cord) may potentially result in one other form of the disease, that is, NT or cerebral toxocariasis [32,165,185]. Autopsy findings of patients with cerebral toxocariasis have demonstrated the presence of parasites in various areas of the CNS, for example, the leptomeninges, the cerebellum, the gray and white matter of the brain, and the spinal cord [32,41,114,186]. Interestingly, in most patients, the clinical picture and necropsy findings did not correlate, given that neurological manifestations were absent in many cases. Therefore, the question arises as to whether the localization of the parasites into the central nervous system is of clinical significance. However, taking into account that between 1951 and 2015, only 100 cases of NT were reported [186] in combination with the high incidence of non-specific symptoms [185,187], an issue that remains open is whether NT is indeed a rare disease or a disease that diverts clinical attention. 

In contrast to VLM, NT is mainly a disease of adulthood, as children younger than 18 years are restrictively affected [186,188]. The clinical spectrum of NT is wide and comprises various clinical entities, for example, meningitis, encephalitis, meningoencephalitis, myelitis, encephalomyelitis, meningomyelitis, and meningoencephalomyelitis. The associated, described clinical symptoms include fever, headache, epileptic seizures, weakness, mental confusion, autonomic dysfunction, sensory and motor disturbances, and paraparesis [9,17,32,41,114,186,189,190,191,192,193,194]. Moreover, accumulating evidence suggests that *Toxocara* infections are causally associated with the development of epilepsy in children and adults [7,195,196,197]; however, taking into account the high prevalence of epilepsy and the high seroprevalence of *Toxocara*, such an association becomes questionable [196,198,199,200,201,202].

Moreover, toxocariasis exhibited a discernible association with an elevated susceptibility to both schizophrenia and bipolar disorders. More specifically, a recent meta-analysis revealed that toxocariasis alone, and the combined occurrence of toxocariasis and toxoplasmosis as well, was linked to an increased susceptibility to schizophrenia. This trend persisted across different subject groups, with hospitalized individuals exhibiting a heightened risk of mental disorders (schizophrenia and/or bipolar disorders). These observations align with the outcomes of a meta-analysis exploring the association between *Toxocara* infection/exposure and schizophrenia, which found a significant link between higher seroprevalence of *Toxocara*- infection/exposure and schizophrenia patients. Collectively, these findings highlight the potential yet often overlooked contribution of *Toxocara* infection/exposure to the development of schizophrenia [203,204].

Blood eosinophilia is not an indispensable finding; it presented in approximately 65% of the patients, whereas IgE appears universally increased [187]. Cerebrospinal fluid (CSF) pleocytosis has been observed in 40 to 64% of the patients [186,187] with eosinophilic predominance. Other CSF findings include increased protein concentration and hypoglycorrhachia [186].

Imaging findings on CT or magnetic resonance imaging [190,205,206] are not pathognomonic and include single or multiple cortical, subcortical, or white matter hyperintense alterations [187,207]. Homogenous or punctate enhancement after administration of contrast agent, multiple or single ring-enhancing lesions, and focal meningeal contrast enhancement is described in a minority of the published cases [187,208,209]. Typical observations in individuals affected by spinal VLM encompass spinal cord edema and elevated signal intensities. While the imaging manifestations of these lesions often lack specificity, resembling non-neoplastic myelopathy akin to transverse myelitis, distinctive features such as singular lesions, concentrated nodular enhancement along the posterior or posterolateral spinal cord segment, succinct segmental engagement, and the potential for lesion migration may collectively denote characteristic attributes of spinal VLM attributed to *T. canis* [208,209,210].

### 5.4. Covert and Common Toxocarosis

In 1987, Taylor and co-workers proposed, additionally, to visceral larva migrans and ocular toxocariasis, the presence of another form of toxocariasis, namely covert toxocariasis, affecting children and presenting as a combination of several signs and symptoms including pulmonary manifestations, for example, cough, pneumonia, and wheezing, pharyngitis, limb pains, cervical lymphadenitis, abdominal pain, hepatomegaly, vomiting, headaches, lethargy, and sleep and behaviour disturbances. Eosinophilia is not an obligatory laboratory abnormality, as it is observed in only 50% of the cases [132,211].

Common toxocariasis, initially observed in French adults, is characterized by a combination of symptoms, including asthenie, cutaneous manifestations, for example, pruritus and rash, dyspnea, and abdominal pain associated with blood eosinophilia, elevated levels of total serum IgE, and enhanced titers of anti-*Toxocara* antibodies [132].

## 6. Diagnosis of Toxocariasis in Humans

The fact that the human intestine is not colonized by adult worms causes diagnostic problems [212]. As a rule, neither *Toxocara* eggs nor larvae or other parasite stages are excreted in the human stool, which makes stool examination unhelpful. Direct detection of larvae in biopsies or punctates is invasive and, in addition, rarely successful and should be only performed when indicated. In addition, it is difficult to distinguish between larvae of *Toxocara* and other ascarids, especially if the larvae degenerate or if only parts of the larva can be recovered from tissues [213,214,215,216]. In general, the diagnosis of toxocariasis should be based on history, clinical examination, and laboratory findings (leukocytosis and eosinophilia), and can be confirmed by serology [213]. There are several enzyme-linked immunosorbent assays (ELISA) that detect human IgG antibodies to *Toxocara* excretory/secretory antigens (TES) of third-stage larvae of *T. canis*. Reliability varies depending on the test and clinical presentation. For example, the sensitivity and specificity of the excretory–secretory-based ELISA for the detection of IgG antibodies for visceral larva migrans are reported to be 78% and 92%, respectively [36,217,218,219,220,221,222]. It should be noted that the amount and type of antibodies cannot distinguish between acute and chronic infection [223]. The results must be interpreted in conjunction with the clinical symptoms and exposure to *Toxocara*. However, a negative test can exclude VLM [213], although it should be noted that ocular and neurological toxocariasis can occur even with negative serology. A further complicating factor is the cross-reactivity of the ELISA test with other parasite antigens. In addition, the test can remain positive for years after therapy. In unclear cases, positive ELISA tests can be confirmed by Western blot [224,225]. The Western blot, which is more expensive and labor-intensive, has a higher sensitivity and specificity when compared to the ELISA [223,226]. Specific detection of total anti-*Toxocara* IgG antibodies and subclasses (e.g., IgG1, IgG2, IgG3, and IgG4) and IgA antibodies is also possible [227,228]. In addition, IgE- and IgM- based on ELISAs can be used to monitor the effectiveness of treatment [229]. However, it is pertinent to highlight that IgM antibodies lack specificity in the context of serodiagnosis for human toxocariasis [230,231]. 

Regrettably, the precision of the majority of commercially accessible serodiagnostic assays for this affliction remains suboptimal. This concern assumes paramount significance, especially within tropical locales where co-infections involving various helminthic pathogens are endemic. Furthermore, despite serology’s pivotal role in diagnosing this zoonotic disease, diagnostic kits predominantly employ crude excretion/secretion antigens, primarily glycoproteins, lacking species specificity. These antigens may elicit cross-reactivity with antibodies generated against unrelated parasites. In pursuit of an enhanced serological assay for the detection of *Toxocara* infection, an IgG(4)-ELISA based on a recombinant variant (rTES-30USM) of the 30-kDa *Toxocara* excretory–secretory antigen (TES-30) has recently emerged. Data derived from two recent studies demonstrate that this novel IgG(4)-ELISA exhibits high sensitivity and specificity. Consequently, this assay stands as an encouraging choice for implementation in tropical regions and any other areas where potentially cross-reactive helminthic infections are prevalent, and in epidemiological studies as well [232,233].

Molecular techniques have high analytical specificity and shorter turnaround times than other diagnostic methods. Polymerase chain reaction (PCR)-based assays using a variety of genetic markers (e.g., ITS-1 and ITS-2 regions of rDNA) have now been developed and have enabled the identification and phylogenetic analysis of *T. canis*, *T. cati*, and other ascarids [234]. Vitellogenin (Vg), recognized as a reservoir of both amino acids and lipids within the eggs of the parasites, is acknowledged to hold a pivotal role in the embryonic development of a diverse spectrum of organisms. A recent study focused on quantifying the transcription levels of the Tc-vit-6 gene in adult male and female *T. canis* using quantitative real-time PCR revealed a significant transcription of the Tc-vit-6 gene in specific anatomical regions, including the intestines, reproductive tract, and body wall of both male and female *T. canis*, highlighting the multifaceted roles of TcVg6 in *T. canis*, extending beyond its involvement in the reproductive processes of the parasite [235] and making it a potential future point of research in the diagnosis of toxocariasis.

## 7. Treatment

The therapeutic approach to toxocariasis represents a medical challenge. As mentioned above, it should be differentiated between asymptomatic *Toxocara* infection in a seropositive patient without symptoms and toxocariasis (e.g., a manifest infection) [165]. Due to the self-limited nature of the disease, it is suggested that treatment should be evaluated in symptomatic patients with severe manifestations [236,237]. 

Traditionally, the therapy of human toxocariasis has relied on the use of anthelmintics and anti-inflammatory drugs [223]. Anthelmintics, also known as antihelminthics, are a group of therapeutic regimens and chemical compounds broadly used in the treatment of parasitic infections. The group of anthelmintic drugs includes a variety of pharmaceutical agents, for example, benzimidazoles (*albendazole*, *mebendazole*), *ivermectin*, *nitazoxanide*, *praziquantel*, *pyrantel*, *pamoate,* and *niclosamide* [238,239,240,241]. *Albendazole* (ABZ) has been the cornerstone for the first-line treatment of human toxocariasis for many decades [242]. The inclination towards ABZ, compared to other antihelminthic agents, may be attributed to its comparatively safe profile with minimal side effects, widespread availability in most countries, and the lower absorption rate of *mebendazole* (MBZ) beyond the gastrointestinal tract [243,244,245,246]. ABZ is characterized by its high potency against a variety of infections caused by many nematodes and some trematodes, and protozoa [241]. Recently, in a collaborative pursuit to alleviate the impact of parasitic diseases prevalent in developing regions, a study focused on the creation of a streamlined compound, N-(4-methoxyphenyl) pentanamide (N4MP), modeled after albendazole’s structure. This compound was synthesized, and its effectiveness was assessed in vitro against infective third-stage larvae (L3) of *T. canis*. The evaluation encompassed analyses of physicochemical attributes, pharmacokinetics, drug-likeness, and medicinal chemistry compatibility for both albendazole and the simplified derivative, N4MP. Comparable to albendazole, bioassays revealed N4MP’s concentration- and time-dependent impact on parasite viability. Interestingly, N4MP exhibited lower cytotoxicity to human and animal cell lines than albendazole. Furthermore, pharmacokinetic assessments, drug-likeness evaluations, and medicinal chemistry assessments collectively portrayed N4MP as possessing an excellent drug-likeness profile, aligned with major pharmaceutical standards. Altogether, these findings highlight the potential of molecular simplification, exemplified by N4MP, as a promising avenue in the exploration of novel anthelmintic agents [242].

Although the World Health Organization (WHO) suggests that women of reproductive age, including pregnant women after the first trimester, should be included in large-scale deworming programs, given the teratogenic effects exhibited in experimental studies according to the safety of ABZ in pregnant women and the existed insufficient evidence, the use of antihelminthic drugs for pregnant women cannot be recommended. Future research in pregnant individuals should focus on presenting information disaggregated by trimester and demonstrating data on maternal and child adverse events, where this became possible [247,248,249]. 

### 7.1. Visceral Larval Migrans

ABZ, at 400 mg twice a day for five days, is the current recommended dose for adults [213]. However, doses of 10 up to 15 mg/kg/day taken orally twice daily have also been reported with sufficient therapeutic rates [86,246]. In some patients with visceral disease, a repetition of the treatment may be necessary. Treatment response could be assessed by clinical investigation and the observation of improvement in eosinophilia and serological tests performed over a time period of at least four weeks [86]. Even though side effects of the therapy (e.g., constipation, diarrhea, vomiting, epigastric pain, lethargy, headache, and leucopenia) are usually mild [250], severe drug-associated reactions may also occur. These include hepatotoxicity, commonly induced by higher doses of ABZ, neurologic manifestations, and hematologic abnormalities, such as cytopenias. It should be noted, however, that in predisposed individuals, even therapeutic doses of ABZ may cause liver toxicity in both adults and children [251,252,253,254,255,256]. Serious, underlying diseases, such as immunosuppression, have been a cause of death under treatment with ABZ, but this is extremely rare [250].

MBZ is an alternative treatment option, with low extraintestinal absorption and rapid first-pass metabolism as major disadvantages [132,213,257]. Traditionally, it has been used as a second-line therapy [132,213]. Three randomized trials have investigated the efficacy of MBZ in treating common/covert toxocariasis. In these studies, MBZ was administrated at 25 mg/kg b/w daily for seven days [245] or 20–25 mg/kg b/w daily for three weeks [244]. In a double-blind, placebo-controlled, randomized study investigating the efficacy of MBZ versus placebo, Magnaval and Charlet used a discontinuous regimen (10–15 mg/kg b/w daily for three consecutive days weekly for six weeks) and found similar efficacy of MBZ versus placebo [258]. However, continuous administration at higher doses resulted in a greater cure rate, control of clinical symptoms, and normalization of laboratory abnormalities, particularly blood eosinophilia [244,245]. Similarly to ABZ, MBZ has mild side effects, for example, skin abscess, granulocytopenia, arthritis, pruritus, and alopecia. However, in high doses, it can cause anemia and liver damage [257,259,260,261].

Other anthelmintic agents, such as *ivermectin*, should be avoided in the treatment of human toxocariasis due to their very low cure rate [243,262,263].

While the recommendations for treating other forms of VLM are clear, the evidence regarding cardiac toxocariasis is restricted and limited to case reports. In a systematic review, Kuenzli et al. 2016 analyzed 24 cases of cardiac toxocariasis. Given the potentially lethal nature of the disease and based on the fact that in cases who received therapy for 2 to 3 weeks, relapse was observed; the authors suggested higher doses of ABZ (15 mg/kg or 800–1000 mg/day) for a duration of 3 to 4 weeks. Moreover, to avoid severe dose-dependent side effects—specifically, hepatotoxicity and myelosuppression—prolonged therapy (e.g., for more than 3–4 weeks) should be carefully evaluated [112]. Based on the pathophysiology of heart disease representing a combination between direct larval toxicity and damage through activated immunological and hypersensitivity mechanisms, which can be potentially exacerbated after a successful treatment due to the release of antigens of destroyed larvae, the additional administration of corticosteroids reflects a meaningful consideration. However, their therapeutic relevance remains poorly understood and needs further evaluation. The administration of corticosteroids should be evaluated in patients with inflammatory reactions and is associated with a rapid resolution of clinical signs and symptoms. The suggested dose is 1 mg/kg body weight for one week, which should be subsequently tapered according to the clinical course and laboratory parameters (e.g., blood eosinophilia) [112].

### 7.2. Ocular Toxocarosis

The diagnosis of OT represents a diagnostic challenge, something that may explain the limited existing data regarding the therapy of OT [246]. Even though some published cases of OT showed a favorable outcome when anthelmintic drugs were combined with corticosteroids; the role of such drugs, for example, ABZ, remains unclear [246,264,265,266,267]. However, the therapy of OT is based on anthelmintic drugs in combination with corticosteroids or surgical interventions, depending on previous ocular comorbidities and grade of inflammation [268,269,270]. In a retrospective study of OT presenting with uveitis and retinochoroidal granulomas, Barisani–Asenbauer and co-authors (2001) showed the resolve of the inflammatory process and elimination of the granulomas in patients who received 800 mg ABZ daily for two weeks combined with corticosteroids [113]. Systemic or topical corticosteroids are used to control intraocular inflammation and related symptoms, but their potency to resolve structural retinal complications is limited [264]. Surgical treatment may be necessary in cases with an epiretinal membrane, persistent vitreous opacity, and retinal detachment [152,178,179,184,271,272,273]. Alternative treatments, such as cryotherapy [274,275] and photocoagulation [276,277], may be considerable therapy modalities in specific patients; however, more research in this direction is necessary.

### 7.3. Neurotoxocariasis

Unfortunately, due to the rarity of NT [186], no controlled studies have taken place in this direction. However, knowledge of other parasitic infections of the CNS, for example, neurocysticercosis, indicates that combining ABZ with corticosteroids may represent an effective therapeutic option [278]. The duration of therapy should be at least three weeks. In some cases, repetition may be needed and is facilitated by the efficient penetration of ABZ into the CNS with no significant toxicity [279,280]. Moreover, pharmacokinetic studies have shown that corticosteroids elevate the concentrations of ABZ in plasma by 50% [281]. The suggested dose of ABZ is 10–15 mg/kg daily, which should be administered until complete resolution of the clinical symptoms and normalization of the MRI, generally over a time period of 21 to 28 days [213,279,282,283]. 

Although there are no recommendations regarding the utility of corticosteroids in the treatment of NT, about 90% of the patients receive corticosteroids in combination with antihelminthic medication to eliminate inflammation and control hypersensitivity/immunological reactions associated with the degeneration of larvae following the treatment of NT. Moreover, some patients with myelitis have been treated with corticosteroid-monotherapy. In the vast majority of cases, a clinical and radiological improvement is reported, including patients with spinal cord lesions managed with corticosteroids alone [186,187,210,284].

## 8. Prevention

Despite the fact that notable progress has been achieved in the prevention, control, and elimination of zoonotic diseases [285], the remarkable expansion of the dog and cat population and, particularly feral and stray animals, contributes to the increased risk of human infection with *Toxocara* in combination with other factors, for example, climate changes and massive migration [285,286,287]. In the Special Issue regarding a United States (US) perspective on significant pet-related parasitic diseases, Paul and coauthors (2010) refer to synergistic work between various disciplines to maintain human, animal, and environmental health as one health approach [288]. Until today there was an effective way to eradicate environmental contamination with *Toxocara* [289]; toxocariasis, however, is preventable through environmental and ecological modifications. Primary prevention goals are the reduction of the parasitic load and the risk of zoonotic transmission [290,291]. A significant contribution in this direction is raising clinicians’ awareness regarding the signs and symptoms of the disease, as well as the education and behavioral changes of the general population on how to avoid parasitic infections. In addition, the role of veterinarians in preventing zoonotic diseases should be stressed. Deworming pets regularly, particularly new kittens and puppies, even when asymptomatic, represents one of the main preventive measures. Moreover, waste deposits in areas in which children play should be strongly avoided [37]. Furthermore, weight must be given to educating the general population regarding the collection and hygienic disposition of pet feces before the eggs become infective [292]. In addition, collective actions regarding stray dogs and cats with deworming at 2–3 weeks of age and two times a year for adults, as well as municipal orders to prevent pet dogs from entering parks and playgrounds, may also contribute to the elimination of the disease [186]. Modifications in human behavioral characteristics, that is, hand hygiene, careful washing of vegetables, and avoiding raw or undercooked meat, represent further preventive challenges [186]. 

## 9. Limitations

Despite the thoroughness of our study, several limitations should be acknowledged when interpreting the results. One limitation of this study is that we did not independently evaluate all included manuscripts for risk of bias. While we tried to minimize bias by following established guidelines and utilizing standardized assessment tools, the potential for bias introduced by not independently assessing each manuscript’s risk of bias can be partially ruled out. This limitation may impact the overall robustness of our findings and the confidence with which we interpret the results. Moreover, our study primarily focused on specific aspects of toxocariasis, which may limit the generalizability of our findings to all subtypes and forms of the disease. Caution is advised when applying our results to less-explored toxocariasis variations and regional contexts.

Another notable area for improvement is the absence of a meta-analysis or statistical synthesis of the available data. While we conducted a comprehensive review of the literature, the heterogeneity in study methodologies, participant characteristics, and the multitude of outcome measures across the selected studies precluded the possibility of conducting a meaningful meta-analysis. Another limitation is the inclusion and use of narrative reviews as a supplementary source of epidemiologic data, as was decided upon to include all meaningful data and expert opinions on the subject, but is a source of bias due to each author’s interpretation and selection of sources. Finally, an inherent limitation of this study lies in excluding a substantial number of primary epidemiological studies, which could have enriched our analysis but were beyond the scope of this research that focused primarily on pathophysiological and clinical aspects of the disease. To address this limitation, we carefully selected, evaluated, and presented data from previous literature reviews and meta-analyses that have employed rigorous methodology to summarize available epidemiologic data.

## 10. Conclusions

Even though toxocariasis represents one of the most common zoonoses worldwide, its diagnosis remains a challenge due to the unfamiliarity of clinicians with its wide clinical spectrum, its diagnostic approaches, and the available treatment options, especially in non-endemic areas. In this work, we aim to address common questions about the aetiology, epidemiology and transmission, clinical manifestations, laboratory and radiologic findings, diagnosis, treatment, and prevention of infections caused by *Toxocara* spp. The disease should be suspected in patients presenting with organ involvement, for example, liver, lungs, and heart, accompanied by blood eosinophilia and elevated IgE, with or without cutaneous manifestations. Serology assays are considered the most useful method for the detection of toxocariasis in patients with suspected clinical disease. In symptomatic patients, anthelmintic drugs such as ABZ, with or without corticosteroids, are the recommended treatment of choice. Prevention strategies should involve multiple disciplines and should include educational programs, behavioral and hygienic changes, enhancement of the role of veterinarians, and municipal orders. Further research is needed in order to better understand the pathophysiology and natural history of the disease, especially in less common or less recognized forms, such as common or covert toxocarosis, OT, and NT. Moreover, there are several gaps in our knowledge that follow from our findings and would benefit from further research, including our understanding of the epidemiology of this parasitic disease, its impact on the pathophysiology of various systematic diseases, such as neurodegenerative disorders and asthma, and the management of this common zoonosis during pregnancy to sustainably address this neglected parasitic disorder that affects millions of people worldwide.

## Figures and Tables

**Figure 1 ijerph-20-06972-f001:**
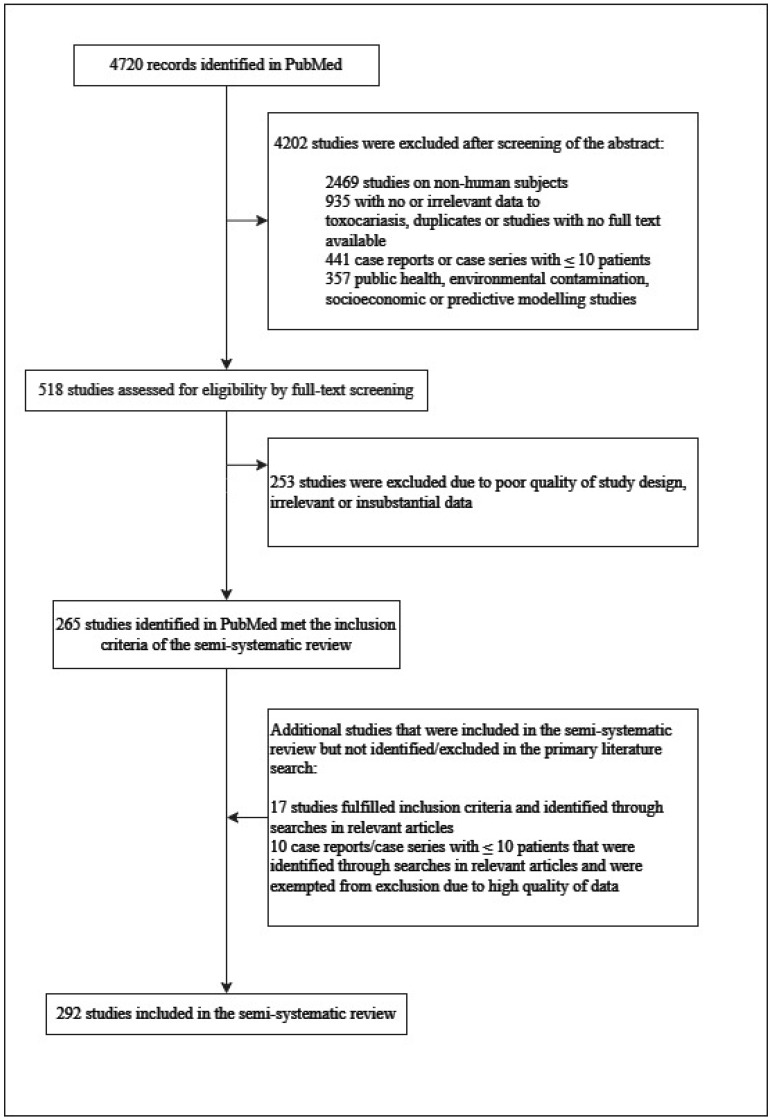
Flowchart illustrating the number (292) of the patients included.

**Figure 2 ijerph-20-06972-f002:**
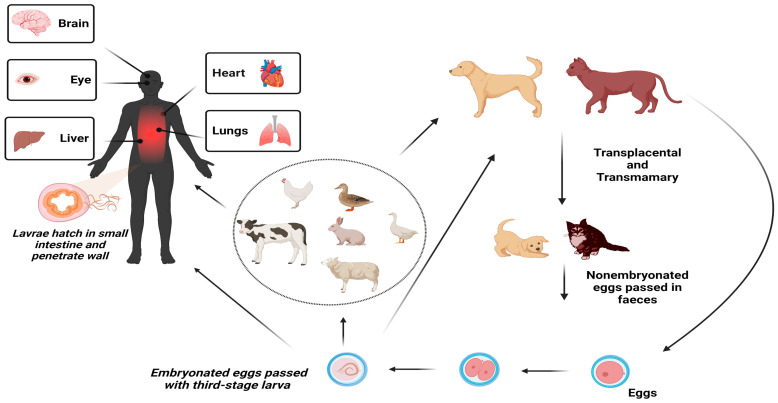
*Toxocara* lifecycle.

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
