# Peer review of "Who Let the Dogs Out? Unmasking the Neglected: A Semi-Systematic Review on the Enduring Impact of Toxocariasis, a Prevalent Zoonotic Infection"

_ijerph, 2023, doi:10.3390/ijerph20216972_

Round 1

Reviewer 1 Report (Previous Reviewer 4)

Comments and Suggestions for Authors

The manuscript has been revised nicely and can be accepted for publication. 

Comments on the Quality of English Language

Minor editing of language. 

Author Response

Reviewer 1

Comment 1: The manuscript has been revised nicely and can be accepted for publication.

Reply: Thank you for your positive feedback and for considering our revised manuscript for publication. We sincerely appreciate your time, effort, and expertise in reviewing our work.

Reviewer 2 Report (New Reviewer)

Comments and Suggestions for Authors Author Expertise: Prioritizing reviews authored by experts in the field is a sensible approach to ensure credibility and reliability. It's essential to provide clarity on how author expertise was determined.   Quality Assessment: The mention of subjecting selected narrative reviews to critical appraisal for methodological rigor and potential biases is commendable. However, it would be beneficial to specify the criteria used for this assessment to enhance transparency.

Scope and Generalizability: Discuss the scope of your study and how it may impact the generalizability of your findings. Be transparent about the specific aspects of toxocariasis that your research focused on and acknowledge that your results may not cover the entire spectrum of the disease.

Author Response

Reviewer 2

Comment 1: Author Expertise: Prioritizing reviews authored by experts in the field is a sensible approach to ensure credibility and reliability. It's essential to provide clarity on how author expertise was determined.  

Reply: Thank you for your comment. In response to your suggestion, we have made the necessary adjustments to define the authors’ expertise.

Comment 2: Quality Assessment: The mention of subjecting selected narrative reviews to critical appraisal for methodological rigor and potential biases is commendable. However, it would be beneficial to specify the criteria used for this assessment to enhance transparency.

Reply: Thank you for your recommendation. In response to your comment, we have further enhanced transparency in our methodology. To address this, we have detailed the specific criteria we employed to assess methodological rigor and identify potential biases. This addition will provide readers with a clearer understanding of our approach and ensure greater transparency in our research. 

Comment 3: Scope and Generalizability: Discuss the scope of your study and how it may impact the generalizability of your findings. Be transparent about the specific aspects of toxocariasis that your research focused on and acknowledge that your results may not cover the entire spectrum of the disease.

Reply: Thank you for your comment. Regarding your suggestion to discuss the scope and generalizability of our study, we would appreciate it if we could take a slightly different approach. Instead of including a separate discussion on this matter, we have incorporated a limitation into our study that explicitly acknowledges the boundaries and potential limitations concerning the scope and generalizability. This adjustment, we believe, will address your concern while ensuring the manuscript remains engaging for our readers. In case of disagreement, we are prepared to discuss the scope of our study and how it may impact the generalizability accordingly.

This manuscript is a resubmission of an earlier submission. The following is a list of the peer review reports and author responses from that submission.

Round 1

Reviewer 1 Report

Comments and Suggestions for Authors

General

The review is not sufficiently contributive in the field in comparison with some other recent reviews.

The references are often old and some important recent references are missing as the following : Healy SR, Morgan ER, Prada JM, Betson M (2022). Brain food: rethinking food-borne toxocariasis. Parasitology 149, 19.

The written is not optimal in its structure and sometimes repetitive.

The pathophysiology section is not very interesting and is a little confusing.

In the Conclusion section, it could be interesting to identify the knowledge gaps in our understanding of toxocarosis.

Specific :

The toxocara life cycle is bad. It suggests that the third-stage larva is the only infective stage for Human and it does not show the real complexity of the cycle in dogs and cats. Paratenic hosts do not appear either on this life cycle.

Line 101 : « toxocariasis ». Authors explain above that the correct term is toxocarosis.

Line 107 : delete « for example »

Group lines 107-108 with lines 111-113.

Line 138 delete « radicals »

The sentence lines 139-142 is unclear.

Line 315-316 : « the sensitivity and specificity are between 78% and 92% » I am not sure of the meaning of this sentence.

Line 319 : « a negative test can exclude VLM ». A reference should be given to document this point.

Line 345 : it should be noted that albendazole is contraindicated in pregnant women.

Comments on the Quality of English Language

The quality of the English langage is correct

Author Response

General

Comment 1 : The review is not sufficiently contributive in the field in comparison with some other recent reviews.

Reply: We understand your concern regarding the contribution of our review compared to other recent reviews. We acknowledge that there are several existing articles on the topic. However, we believe that our work adds value by synthesizing and presenting the available literature comprehensively and cohesively. Thus, we believe that our article makes a valuable contribution to the scientific community by consolidating and presenting the current state of knowledge on the topic and it will be of interest to researchers and professionals working in this field. Once again, we appreciate your feedback and have carefully considered your comments while revising the manuscript.

Comment 2: The references are often old and some important recent references are missing as the following : Healy SR, Morgan ER, Prada JM, Betson M (2022). Brain food: rethinking food-borne toxocariasis. Parasitology 149, 1–9.

Reply: We thank the reviewer for bringing the above issue to our attention. We have now updated our references.

Comment 3 : The written is not optimal in its structure and sometimes repetitive.

Reply: Thank you for your recommendation. We have revised our manuscript accordingly in order to avoid repetitions and enhance readability.

Comment 4 : The pathophysiology section is not very interesting and is a little confusing.

Reply: We thank the reviewer for the very important comment. In light of the edifying recommendation, we have restructured Pathophysiology Section. Moreover, and in order to the suggestion of Reviewer 4, we have added some additional data regarding the role of eosinophils in the Pathophysiology of the disease. Kindly see also the response to Reviewer 4.

Comment 5 : In the Conclusion section, it could be interesting to identify the knowledge gaps in our understanding of toxocarosis.

Reply: Thank you for your comment. Future areas of research are now stated in the “Conclusions” section.

Specific :

Comment 6 : The toxocara life cycle is bad. It suggests that the third-stage larva is the only infective stage for Human and it does not show the real complexity of the cycle in dogs and cats. Paratenic hosts do not appear either on this life cycle.

Reply: We have rewritten Toxocara’s Life Cycle to avoid inconsistencies and ambiguities.

Comment 7 : Line 101 : « toxocariasis ». Authors explain above that the correct term is toxocarosis.

Reply: Thank you for your observation. However, in order of the suggestions of Reviewer 2 and 4 we have now used the term “toxocariasis”  to be consistent with most English publications.

Comment 8: Line 107 : delete « for example »

Reply: Done as recommended.

Comment 9: Group lines 107-108 with lines 111-113.

Reply: Done as recommended.

Comment 10: Line 138 delete « radicals »

Reply: Done as recommended.

Comment 11: The sentence lines 139-142 is unclear.

Reply: We have rephrased the sentence lines 139-142.

Comment 12: Line 315-316 : « the sensitivity and specificity are between 78% and 92% » I am not sure of the meaning of this sentence.

Reply: We agree with the Reviewer. We have now rewritten the sentence.

Comment 13: Line 319 : « a negative test can exclude VLM ». A reference should be given to document this point.

Reply: Done as recommended.

Comment 14: Line 345 : it should be noted that albendazole is contraindicated in pregnant women.

Reply: Done as recommended.

Reviewer 2 Report

Comments and Suggestions for Authors

Comments to the Authors

Undoubtedly, human toxocarosis is an important medical issue and any research on the topic should be appreciated.

However, I have some minor and a few major remarks about the text.

Minor remarks:

-1- Line 78 – Probably it should be mentioned that in the literature in English the most common term is “toxocariasis”.

-2- Lines 138, 160, 352, 442 and 447 contain some technical errors that should be corrected.

Major remarks:

-1- The Introduction section should be significantly shortened and rewritten. Much of the current content repeats the content of the subsequent sections and subsections, which is not a good practice in the scientific literature.

-2- About the cutaneous larva migrans, from Line 227 to Line 238: I kindly will ask the Authors to read again the references they give in this little subsection, namely, references 102, 103, 104, 105, 106 and 107. Reading references 102, 103, 104 and 105, the Authors will understand that cutaneous larva migrans is caused by animal hookworms such as Ancylostoma braziliense. Ancylostoma caninum, Ancylostoma ceylonicum, Ancylostoma tubaeforme, Strongyloides papillosus, Uncinaria stenocephala and others, and that they all are able to penetrate the human skin from the outside environment (most often soil) and cause creeping eruption.

Reading again references 106 and 107, the Authors will see that the cutaneous manifestations of human toxocariasis/toxocarosis most often are pruritus, urticaria, atopic dermatitis etc. The pathophysiology of the cutaneous manifestations of human toxocariasis/toxocarosis is completely different from these in cutaneous larva migrans and the Toxocara larvae do not have to be even located in the skin. Moreover, the Toxocara larvae never ever penetrates the human skin from the outside environment.

Therefore, there is nothing in common between Visceral Larva Migrans and Cutaneous Larva Migrans, except that the causative agents are nematodes.

Author Response

Comments to the Authors

Undoubtedly, human toxocarosis is an important medical issue and any research on the topic should be appreciated.

However, I have some minor and a few major remarks about the text.

Minor remarks:

Comment 1: Line 78 – Probably it should be mentioned that in the literature in English the most common term is “toxocariasis”.

Reply: Done as recommended.

Comment 2: Lines 138, 160, 352, 442 and 447 contain some technical errors that should be corrected.

Reply: Done as recommended.

Major remarks:

Comment 3: The Introduction section should be significantly shortened and rewritten. Much of the current content repeats the content of the subsequent sections and subsections, which is not a good practice in the scientific literature.

Reply: Done as recommended.

Comment 4: About the cutaneous larva migrans, from Line 227 to Line 238: I kindly will ask the Authors to read again the references they give in this little subsection, namely, references 102, 103, 104, 105, 106 and 107. Reading references 102, 103, 104 and 105, the Authors will understand that cutaneous larva migrans is caused by animal hookworms such as Ancylostoma braziliense. Ancylostoma caninum, Ancylostoma ceylonicum, Ancylostoma tubaeforme, Strongyloides papillosus, Uncinaria stenocephala and others, and that they all are able to penetrate the human skin from the outside environment (most often soil) and cause creeping eruption.

Reading again references 106 and 107, the Authors will see that the cutaneous manifestations of human toxocariasis/toxocarosis most often are pruritus, urticaria, atopic dermatitis etc. The pathophysiology of the cutaneous manifestations of human toxocariasis/toxocarosis is completely different from these in cutaneous larva migrans and the Toxocara larvae do not have to be even located in the skin. Moreover, the Toxocara larvae never ever penetrates the human skin from the outside environment.

Therefore, there is nothing in common between Visceral Larva Migrans and Cutaneous Larva Migrans, except that the causative agents are nematodes.

Reply: We thank the reviewer for the very important comment. In light of the edifying recommendation, we have rewritten the “Cutaneous Manifestations” and corrected ambiguities.

Reviewer 3 Report

Comments and Suggestions for Authors

This manuscript describes the common questions about the aetiology, epidemiology and transmission, clinical manifestations, laboratory and radiologic findings, diagnosis, treatment and, prevention of infections caused by Toxocara spp. Although it is good review of the biology of Toxocara spp. based on the different literature with a certain theme for a comprehensive overview, there is a certain reference value, but also a simple list of the overview together, it did not point out the future direction of research, the future research hot spots, which direction should be developed? the existence of deficiencies, and so on.

Please note the spaces in the full text, the letter case of words; the writing of parasite names, when it appears for the first time, use the full name (Toxocara canis; Toxocara cati) (abbreviated), thereafter all use abbreviations (T. canis; T. cati)

In the references, please note the case of words!

Please note the yellow markers.

Please follow the Authors Guidelines strictly and carefully.

Author Response

Comment 1: This manuscript describes the common questions about the aetiology, epidemiology and transmission, clinical manifestations, laboratory and radiologic findings, diagnosis, treatment and, prevention of infections caused by Toxocara spp. Although it is good review of the biology of Toxocara spp. based on the different literature with a certain theme for a comprehensive overview, there is a certain reference value, but also a simple list of the overview together, it did not point out the future direction of research, the future research hot spots, which direction should be developed? the existence of deficiencies, and so on.

Reply: Thank you for your comment. Existing research gaps and suggestions for future research are now referred to in the manuscript and in the “Conclusions” section as well.

Comment 2: Please note the spaces in the full text, the letter case of words; the writing of parasite names, when it appears for the first time, use the full name (Toxocara canis; Toxocara cati) (abbreviated), thereafter all use abbreviations (T. canis; T. cati)

Reply: Done as recommended.

Comment 3: In the references, please note the case of words!

Reply: Thank you for the feedback. We appreciate your attention to detail. We would like to address the concern you raised regarding the case of words in the references. As we used Endnote for managing our references, we kindly request your specific guidance or clarification on the precise issue that needs to be corrected. This will help us to accurately address the matter and ensure that our references meet the required standards.

Comment 4: Please note the yellow markers.

Reply: Done as recommended.

Reviewer 4 Report

Comments and Suggestions for Authors

The manuscript if very interesting and important. zoonotic helminthiasis is a big problem. This manuscript elegantly described toxocariasis covering epidemiology, pathology, immunology, diagnosis and therapy. However, the following points need to be addressed before being accepted for publication-

1. It is better to use 'Toxocariasis' in the title instead of 'Toxocarosis'.

2. Abstract: Is it 1.4 billion; I think over estimated since A. lubricodes, the most common nematode affects only 1.5 bill ( pls see Anisuzzaman et al. “Food- and vector-borne parasitic zoonoses: Global burden and impacts.” Advances in parasitology vol. 120 (2023): 87-136. doi:10.1016/bs.apar.2023.02.001, among others

3. Introduction: Life cycle not properly written since eggs develop in the environment to l2 containing eggs, which is infective.

Paratenic host not included

L2 containing eggs   are infective, not L3 containing eggs

Development of eggs occur in the environment not in the intestine.

The parasite is not most common in humans, the most common nematode is  AL and hookworms. 

93% prevalent?? in human or dogs?? Over estimated. 

Add DALY (pls have an idea from Anisuzzaman et al 2023 Adv Parasitol; Anisuzzaman et al 2020,  Parasitol Int.).

Epidemiology: Please mention 'infective eggs' as the as the agent responsible for infection. Unsegmented eggs can not cause infection

L3 containing eggs are not infective

Really sheep, cattle and chickens have role in the life cycle? Pls check

Fig 1: Pls add paratenic hosts; not complete

Roles of eosinophils have been described in nodular worms in cattle and tick infection in mice. (Pls see Anisuzzaman et al, Parasitol Int, Anisuzzaman et al JCI)

Pls mention the names of some eosinophilopoietic cytokines

Tr4eatment: PZQ and triclabendazole are not for nematodes

MBZ; pls full in the first occurance then abbreviated throughout

Italicize all scientific name. Scientific name full ate first occurance then abbreviated   throughout. 

Comments on the Quality of English Language

Minor checking

Author Response

The manuscript if very interesting and important. zoonotic helminthiasis is a big problem. This manuscript elegantly described toxocariasis covering epidemiology, pathology, immunology, diagnosis and therapy. However, the following points need to be addressed before being accepted for publication-

Reply: Thank you for your comments.

Comment 1: It is better to use 'Toxocariasis' in the title instead of 'Toxocarosis'.

Reply: Done as recommended.

Comment 2: Abstract: Is it 1.4 billion; I think over estimated since A. lubricodes, the most common nematode affects only 1.5 bill ( pls see Anisuzzaman et al. “Food- and vector-borne parasitic zoonoses: Global burden and impacts.” Advances in parasitology vol. 120 (2023): 87-136. doi:10.1016/bs.apar.2023.02.001, among others) 

Reply: We thank the reviewer for bringing the above issue to our attention. However, in order to the suggestion of Reviewer 2, we have shortened and rewritten the Abstract, and this observation is now not stated in the Abstract. However, kindly see References 1 and 2.

Comment 3: Introduction: Life cycle not properly written since eggs develop in the environment to l2 containing eggs, which is infective. Paratenic host not included. L2 containing eggs   are infective, not L3 containing eggs. Development of eggs occur in the environment not in the intestine.

Reply: In light of the edifying comments, we have revised the “Epidemiology Section” accordingly, and these very important observations are now included and clarified.

Comment 7: The parasite is not most common in humans, the most common nematode is  AL and hookworms. 

Reply: Thank you for your comment. In order to the helpful recommendation, we have rephrased it as “Even though the World Health Organization and Centers for Disease Control classified toxocariasis amongst the top six parasitic infections of priority to public health”.

Comment 8: 93% prevalent?? in human or dogs?? Over estimated. 

Reply: We thank the reviewer for the recommendation. We have changed the sentence to “Reported seroprevalence of toxocariasis ranges between 2% and 37% in urban and rural areas of Europe and the USA but may reach 85% in rural tropical regions”. Moreover, we have added an additional reference that refers to the seroprevalence of 85% in rural tropical regions.

Comment 9: Add DALY (pls have an idea from Anisuzzaman et al 2023 Adv Parasitol; Anisuzzaman et al 2020,  Parasitol Int.).

Reply: Done as recommended.

Comment 10: Epidemiology: Please mention 'infective eggs' as the as the agent responsible for infection. Unsegmented eggs can not cause infection. L3 containing eggs are not infective

Reply: Done as recommended.

Comment 12: Really sheep, cattle and chickens have role in the life cycle? Pls check

Reply: Sheep, cattle, and chickens refer to the transmission of the parasite through the consumption of raw or inadequately cooked meat. We have now clarified the inconsistency.

Comment 13: Fig 1: Pls add paratenic hosts; not complete

Reply: Done as recommended.

Comment 14: Roles of eosinophils have been described in nodular worms in cattle and tick infection in mice. (Pls see Anisuzzaman et al, Parasitol Int, Anisuzzaman et al JCI)

Reply: We thank the reviewer for the constructive comment. We agree that the pathophysiology of toxocariasis is characterized by its complexity and that experimental data on the role of eosinophils are not mentioned in detail. Nevertheless, due to the complexity of the pathophysiological mechanisms and the fact that the aforementioned issue could be the subject of a separate review, we aimed to discuss the general and most common pathogenetic mechanisms in order to provide an overview of the pathophysiology of toxocariasis. However, in light of the edifying comment, we have revised the manuscript accordingly and added some very important observations of the suggested references. The very important observation regarding the complexity of the pathophysiology of the diseases is now included in the manuscript.

Comment 15: Pls mention the names of some eosinophilopoietic cytokines

Reply: Done as recommended.

Comment 16: Tr4eatment: PZQ and triclabendazole are not for nematodes

Reply: Thank you for your comment. We have now corrected the inconsistency.

Comment 17: MBZ; pls full in the first occurance then abbreviated throughout

Reply: Done as recommended.

Comment 18: Italicize all scientific name. Scientific name full ate first occurance then abbreviated   throughout. 

Reply: Done as recommended.

Round 2

Reviewer 2 Report

Comments and Suggestions for Authors

As is corected, the review may be of use.

Comments on the Quality of English Language

The English is OK.

Reviewer 3 Report

Comments and Suggestions for Authors

The authors have made a very careful and comprehensive revision. Accept in present form.